# SOX2 Represses Hepatitis B Virus Replication by Binding to the Viral EnhII/Cp and Inhibiting the Promoter Activation

**DOI:** 10.3390/v12030273

**Published:** 2020-02-29

**Authors:** Hua Yang, Jiayin Mo, Qi Xiang, Peiyi Zhao, Yunting Song, Ge Yang, Kailang Wu, Yingle Liu, Weiyong Liu, Jianguo Wu

**Affiliations:** 1State Key Laboratory of Virology, College of Life Sciences, Wuhan University, Wuhan 430072, China; 2014102040012@whu.edu.cn (H.Y.); mojiayin@whu.edu.cn (J.M.); 2017202040043@whu.edu.cn (Q.X.); 2018202040050@whu.edu.cn (P.Z.); 2018202040016@whu.edu.cn (Y.S.); 2016202040041@whu.edu.cn (G.Y.); wukailang@whu.edu.cn (K.W.); mvlwu@whu.edu.cn (Y.L.); 2Guangdong Provincial Key Laboratory of Virology, Institute of Medical Microbiology, Jinan University, Guangzhou 510632, China; 3Department of Clinical Laboratory, Tongji Hospital, Tongji Medical College, Huazhong University of Science and Technology, Wuhan 430030, China

**Keywords:** HBV core promoter, Cp, HBV Enhancer II, EnhII, hepatitis B virus, HBV, sex determining region Y box2, SOX2, viral replication

## Abstract

Hepatitis B virus (HBV) replication is controlled by four promoters (preS1, preS2, Cp, and Xp) and two enhancers (EnhI and EnhII). EnhII stimulates Cp activity to regulate the transcriptions of precore, core, polymerase, and pregenomic RNAs, and therefore, EnhII/Cp is essential for the regulation of HBV replication. This study revealed a distinct mechanism underlying the suppression of EnhII/Cp activation and HBV replication. On the one hand, the sex determining region Y box2 (SOX2), a transcription factor, is induced by HBV. On the other hand, SOX2, in turn, represses the expression levels of HBV RNAs, HBV core-associated DNA, hepatitis B surface antigen (HBsAg), and hepatitis B e antigen (HBeAg), thereby playing an inhibitory role during HBV replication. Further studies indicated that SOX2 bound to the EnhII/Cp DNA and repressed the promoter activation. With the deletion of the high mobility group (HMG) domain, SOX2 loses the ability to repress EnhII/Cp activation, viral RNA transcription, HBV core-associated DNA replication, HBsAg and HBeAg production, as well as fails to enter the nucleus, demonstrating that the HMG domain is required for the SOX2-mediated repression of HBV replication. Moreover, SOX2 represses HBsAg and HBeAg secretion in BALB/c mice sera, and attenuates HBV 3.5 kb RNA transcription and hepatitis B virus core protein (HBc) production in the liver tissues, demonstrating that SOX2 suppresses HBV replication in mice. Furthermore, the results revealed that the HMG domain was required for SOX2-mediated repression of HBV replication in the mice. Taken together, the above facts indicate that SOX2 acts as a new host restriction factor to repress HBV replication by binding to the viral EnhII/Cp and inhibiting the promoter activation through the HMG domain.

## 1. Introduction

Hepatitis B virus (HBV) had infected about 3 billion people worldwide, which led to 800,000 deaths per year [1,2]. When a person is first infected with HBV, it is classified as an acute infection. If the infection remains for over six months, it is said to be a chronic infection (lifelong) [3]. Thus, HBV infection is the most common chronic viral infection in the world. High morbidity and mortality are attributed to chronic infection with HBV, carrying a high risk of progression into liver cirrhosis (LC) and hepatocellular carcinoma (HCC) [4]. The HBV genome contains a partially double-stranded 3.2-kb DNA [5]. Upon infection, the circular DNA (rcDNA), relaxed by HBV, is converted into a covalently closed circular DNA (cccDNA) in the nucleus, which acts as the template for pregenomic RNA (pgRNA), subgenomic precore RNA, 2.4 kb RNA, 2.1 kb RNA, and 0.7 kb RNA [6,7]. The pgRNA serves as the template for reverse transcription, HBV polymerase and HBcAg. HBV pgRNA acts as the template to synthesis progeny virus rcDNA. Viral genes are transcribed under the regulation of four promoters (preS1, preS2, core, and X) and two enhancers (Enhancer Ι and Enhancer ΙΙ). The core promoter is responsible for the synthesis of pgRNA and the regulation of the core promoter is important in the replication and life cycle of HBV [8]. Enhancer I regulates Xp and other viral promoters. Enhancer II stimulates the transcriptional activity of HBV precore/core promoter, which regulates the transcriptions of precore, core, polymerase, and pgRNA [9]. 

SOX2, which belongs to sex-determining region of the Y chromosome (SRY)-related high mobility group-box gene (SOX) family, is a critical transcription regulator for cell proliferation and tumorigenicity of solid tumors, such as glioblastoma, breast cancer, pancreatic carcinoma, lung, and esophageal squamous cell carcinomas [10]. The expression level of SOX2 decreases in mouse liver tissues and in human PBMCs with ageing [11]. SOX2 is also involved in the progression of HCC. For instance, SOX2 is a main downstream regulator of the silent information regulator 1 (SIRT1)-mediated self-renewal and tumorigenicity of liver cancer stem cells [12]. Moreover, overexpression of SOX2 and its co-factor Oct4 can be novel predictors of poor prognosis for patients undergoing resection of HCC [10]. SOX2 activates the expression of programmed death ligand-1 (PD-L1) through direct binding to the PD-L1 promoter in hepatoma cells. Binding of PD-L1 to its receptor, programmed death-1 (PD-1), leads to tumor evasion from the host immune system [13]. SOX2 is also targeted by miR-126, which can inhibit cell proliferation in HCC cells [14]. In a previous study, it was revealed that HBV could induce the production of cancer stem cells (CSCs)-related markers (CD133, CD117, and CD90) and CSCs-related genes (Klf4, SOX2, Nanog, c-Myc, and Oct4) and facilitate the self-renewal of CSCs in human hepatoma cells [15]. The level of SOX2 in normal liver is much lower comparing to that in HBV and alpha fetoprotein (AFP)-positive HCC tissues. Further, SOX2, Klf4, and c-myc were increased by HBx in normal liver cell L-02 and Chang liver [16]. However, the role of SOX2 in the regulation of HBV infection has not been reported. 

This study revealed that SOX2 played an inhibitory role during HBV replication by repressing HBV RNAs, HBV core-associated DNA, hepatitis B surface antigen (HBsAg), and hepatitis B e antigen (HBeAg). Further studies demonstrated that SOX2 bound to the EnhII/Cp DNA and repressed the promoter activation through the high mobility group (HMG) domain, thereby suppressing HBV replication. Moreover, we revealed that SOX2 repressed HBV replication in BALB/c mice and the HMG domain was required for SOX2-mediated repression of HBV replication in vivo. Collectively, these results revealed a distinct mechanism underlying SOX2 repressed HBV replication by binding to the viral EnhII/Cp and inhibiting the promoter activation. 

## 2. Materials and Methods

### 2.1. Animal Study

In the HBV hydrodynamic injection (HI) mouse model, the concentration of injected pAAV-HBV plasmid DNA contributes to the diverse kinetics of HBsAg and HBeAg in the serum as well as HBcAg expression level in liver tissues, which is a useful tool for HBV-related research in vivo [17]. For the in vivo experiments, 5-week-old BALB/c mice were divided into 3 groups (8 for each). Quantities of 10 µg pAAV-HBV1.3 and 10 µg expression vector were co-delivered to the tail veins of mice together. The expression of HBsAg and HBeAg in serum was measured by ELISA at 2 days and 4 days post-injection. Liver tissues were homogenized in the TRIzol reagent and total RNA was purified following the manufacturer’s protocol. A section of 3.5 kb mRNA was detected by qRT-PCR, M-GAPDH and was used as the internal control in qRT-PCR. m-GAPDH forward primers: 5′-TTCACCACCATGGAGAAGGC-3′. m-GAPDH reserve primer: 5′-GGCATCGACTGTGGTCATGA-3′. HBV 3.5 kb RNA forward primer: 5′-GAGTGTGGATTCGCACTCC-3′, HBV 3.5 kb RNA reverse primer: 5′-GAGGCGAGGGAGTTCTTCT-3′. Mouse monoclonal HBcAg (sc-23947) antibody was purchased from Santa Cruz (Dallas, Texas, USA.). Rabbit monoclonal SOX2 (23064) antibody was purchased from Cell Signaling Technology (Danvers, Colorado, USA). All mice were housed in a pathogen-free mouse colony and the animal protocol was performed according to the Guide for the Care and Use of Medical Laboratory Animals (Ministry of Health, China, 1998). The ethical permission code: WDSKY0201901; the permission date: 27 February 2019; and the name of the ethics committee: Ke Lan, Zishu Pan, Ling Zheng, Mingzhou Chen, Yingliang Wu, Xiangdong Gao, and Qian Liu.

### 2.2. Plasmids and Constructions

pHBV1.3 (genotype D, serotype ayw, U9551) was created in our lab. pGL-3-preS1-Luc, pGL-3-preS2-Luc, pGL-3-EnhII/Cp-Luc and pGL-3-EnhI/Xp-Luc were provided by Professor Deyin Guo of Wuhan University. The coding regions of SOX2 (gene ID: 6657) were generated by PCR and inserted into the BamHI and EcoRI sites of pcDNA3.1-3×Flag, which contains a 3×Flag tag. The wild-type and truncated forms of SOX2 were inserted into pcDNA3.1-3×Flag. 

### 2.3. Cell Culture and Transfection

Fetal bovine serum (FBS), Dulbecco’s modified Eagle medium (DMEM) and penicillin/streptomycin were purchased from Gibco (GrandIsland, State of New York, USA).The hepatocarcinoma cell lines HepG2, Huh7, HepG2.2.15 cells (derived from HepG2 cells carrying HBV genome) and HepAD38 (replicates HBV under conditions regulated with tetracycline) were obtained from China Center for Type Culture Collection (CCTCC) and grown in DMEM supplemented with 10% FBS, 100 U/mL penicillin, and 100 µg/mL streptomycin. The cells were seeded in 24-well (seeding densities: 2.5 × 10^5^ cells per well), 12-well (seeding densities: 5 × 10^5^ cells per well), 6-well (seeding densities: 1.2 × 10^6^ cells per well), and 6 cm vessels (seeding densities: 2.6 × 10^6^ cells per dish), and were transfected by Lipofectamine 2000 transfection regent following the manufacturer’s instructions.

### 2.4. Quantitative RT-PCR Analysis

Total RNA was extracted with an Ultrapure RNA Kit (CW0581), which was purchased from Cwbiotech (Beijing, China), according to the manufacturer’s instructions. A quantity of 5 µg RNA was used to synthesize cDNA by using random prime. Then, 2 µL cDNA was detected by a Light Cycler 480 instrument (Roche, Basel, Switzerland). Glyceraldehyde-3-phosphate dehydrogenase (GAPDH) is a house-keeping glycolysis gene and expressed in almost every tissue. GAPDH was used as the internal control. The SOX2 forward primer: 5′-GCCCTGCAGTACAACTCCAT-3′, SOX2 reverse primer: 5′-GACTTGACCACCGAACCCAT-3′. GAPDH forward primer: 5′-GGAAGGTGAAGGTCGGAGTCAACGG-3′, GAPDH reverse primer: 5′-CTCGCTCCTGGAAGATGGTGATGGG-3′. 

### 2.5. Western Blot 

Whole cell protein lysates were resuspended in RIPA buffer supplemented with protease inhibitors, incubated on ice for 30 min, and centrifuged for 10 min at 15,000× *g*. Equal amounts of protein were separated by SDS-PAGE and transferred onto a nitrocellulose (NC) filter membrane (HATF00010, Millipore, Billerica, Massachusetts, USA). The membrane was blocked with skim milk for 1 h and incubated with primary antibodies overnight. Rabbit monoclonal SOX2 (23064) antibody was purchased from Cell Signaling Technology (Danvers, Colorado, USA) (dilution 1:1000). Mouse monoclonal Flag (F1804) antibody was bought from Sigma-Aldrich (Saint Louis, Missouri, USA) (dilution 1:2000). Mouse monoclonal β-actin (60008-1-Ig) antibody (dilution 1:2000), mouse monoclonal GAPDH (60004-1-lg) antibody (dilution 1:2000), and rabbit polyclonal LaminA/C (10298-1-AP) antibody (dilution 1:2000) were purchased from Proteintech Group (Rosemont, Illinois, USA). HRP-conjugated anti-mouse (dilution 1:5000) and anti-rabbit (dilution 1:5000) were used as the secondary antibody. The proteins were detected with ECL (1705060, Bio-Rad, Hercules, California, USA) by LAS-4000 image document instrument (Fujifilm, Tokoy, Japan).

### 2.6. HBV DNA Quantification

HBV core-associated DNA was purified according to the protocol described in our previous study [18]. In brief, the cells were lysed in lysis buffer (50 mM Tris-HCl [pH 7.0], 0.5% NP-40) on ice and centrifuged for 1 min at 10,000× *g*. The supernatant was treated with 10 mM MgCl2 and DNaseI for 1 h at 37 °C. Protein was digested with proteinase K and 1% SDS overnight. HBV core-associated DNA was purified by using Cycle-Pure kit (D6492-01, omega, Wisconsin, USA) and detected by qRT-PCR using HBV qRT-PCR forward primers: 5′-AGAAACAACACATAGCGCCTCAT-3′, HBV qRT-PCR reserve primer: 5′-TGCCCCATGCTGTAGATCTTG-3′. 

### 2.7. Northern Blot

DIG Northern Starter Kit (12039672910) was purchased from Roche (Basel, Switzerland). Total RNA was extracted by Ultrapure RNA Kit and then loaded onto 1.5% agarose gels. The RNAs were transferred onto a nylon membrane. Hybridization was performed following the manufacturer’s instructions containing the DIG-labeled HBV RNA probe. The DIG-labeled probe was detected with LAS-4000 image document instrument. The 28s and 18s rRNAs were used as internal controls to measure the amount of total RNA loading.

### 2.8. Dual-Luciferase Reporter Assay

Cells were seeded in 24-well and transfected with reporter plasmids. The supernatant was removed and cells were lysed by passive lysis buffer at 48 h. Luciferase activity assay was detected by the Dual-Luciferase Reporter Assay System (0000171092, Promega, Madison, Wisconsin, USA). All reporter assays were repeated three times and the data were shown as means ± SD. pRL-TK was used as the transfection efficiency control.

### 2.9. Nuclear and Cytoplasmic Extraction Reagents

NE-PER Nuclear and Cytoplasmic Extraction Reagents (78835) was bought from Promega (Madison, Wisconsin, USA). Cells in 6-well plates were collected after transfection for 48 h. Nuclear and cytoplasmic extraction was purified according to the manufacturer’s instructions. Cocktail protease inhibitors were added to each type of buffer.

### 2.10. Chromatin Immunoprecipitation (ChIP) Assay

Cells were collected and fixed with formaldehyde for 10 min. The glycine at the final concentration of 125 mM was added for 5 min to terminate cross-linking. The precipitated DNA was purified according to the protocol described in our previous study [19]. The DNA was detected by agarose gel electrophoresis and qRT-PCR. ChIP forward primer: 5′-AACGCCCACCAAATATTGCC-3′, ChIP reverse primer: 5′-CTCAAGGTCGGTCGTTGACA-3′.

### 2.11. Immunofluorescence and Confocal Analysis

Cells were grown in glass bottom dishes. At 48 h post-transfection, cells were fixed with 4% paraformaldehyde, 0.2% Triton X-100, 10% BSA, and incubated with anti-Flag (dilution 1:100) overnight at 4 °C. The cells were washed by PBST three times and incubated with Cy3-conjugated goat anti-mouse IgG (dilution 1:100), followed by three times PBST washing. Cells were stained with DAPI and visualized by confocal laser microscopy (Fluoview FV1000; Olympus, Tokyo, Japan).

### 2.12. Statistical Analysis

The results were presented as means ± SD. Student’s t test was performed for statistical comparison between two groups. One-way ANOVO and Tukey: Compare all pairs of columns were performed for statistical comparison between three groups or more. All statistical analyses were performed using Prism 5 software (GraphPad Software Inc. La Jolla, California, USA.). The levels of differences were considered statistically significant at a value of *p* ≤ 0.05. NS, no significant (*p* > 0.05); *, *p* < 0.05, **, *p* < 0.01 and ***, *p* < 0.001. 

## 3. Results

### 3.1. HBV Induces SOX2 Expression

SOX2 mRNA and proteins were significantly higher in HepG2.2.15 cells as compared to that in HepG2 cells (Figure 1A). Similarly, endogenous SOX2 mRNA and proteins were higher in pBlue-HBV1.3 (D)-transfected HepG2 cells as compared to that in pBlue-transfected cells (Figure 1B). Immunohistochemistry showed that SOX2 staining level was significantly higher in the HBV-positive HCC tissues as compared with that in the liver tissues of healthy individual (Figure 1C). Collectively, these results demonstrated that HBV activated SOX2 mRNA and protein expression.

### 3.2. SOX2 Represses HBV Replication in HepG2 Cells and Huh7 Cells

As the effect of SOX2 on the HBV replication had not been reported, we initially investigated the role of SOX2 in the regulation of HBV gene expression and replication. HepG2 and Huh7 cells were co-transfected with pBlue-HBV1.3 (genotype D) and pcDNA3.1-SOX2 at different concentrations. Northern bolt showed that the levels of HBV RNAs (3.5 kb, 2.4 kb, and 2.1 kb RNAs) were attenuated by SOX2 in HepG2 cells (Figure 2A) and Huh7 cells (Figure 2B). Similarly, qRT-PCR indicated that the levels of HBV core-associated DNA were reduced by SOX2 in HepG2 cells (Figure 2C) and Huh7 cells (Figure 2D). Moreover, ELISA assays revealed that hepatitis B surface antigen (HBsAg) and hepatitis B e antigen (HBeAg) in the cell culture supernatants were decreased by SOX2 in HepG2 cells and Huh7 cells in dose-dependent manners (Figure 2E,F). Western blot confirmed that SOX2 protein was produced in pcDNA3.1-SOX2-transfected HepG2 cells (Figure 2G) and Huh7 cells (Figure 2H). Collectively, these results demonstrated that SOX2 played an inhibitory role during HBV replication.

### 3.3. SOX2 Represses HBV Replication through Inhibiting EnhII/Cp Activation

To investigate the mechanisms involved in SOX2-mediated HBV suppression, the role of SOX2 in the regulation of the four HBV promoters (preS1, preS2, EnhII/Cp, and EnhI/Xp) activities were evaluated. HepG2 cells were co-transfected with reporter plasmids, pGL-3-preS1-Luc, pGL-3-preS2-Luc, pGL-3-EnhII/Cp-Luc, or pGL-3-EnhI/Xp-Luc along with pcDNA3.1 or pcDNA3.1-SOX2. The results from the dual-luciferase reporter assay showed that the activity of EnhII/Cp was inhibited by SOX2, whereas the activities of EnhI/Xp, preS1, and preS2 were relatively unaffected by SOX2 in HepG2 cells (Figure 3A). In order to confirm the role of SOX2 in the repression of EnhII/Cp activity, HepG2 cells were co-transfected with pGL-3-EnhII/Cp-Luc and pcDNA3.1 or pcDNA3.1-SOX2 at different concentrations. Dual-luciferase reporter assay revealed that the activity of EnhII/Cp was repressed by SOX2 in a dose-dependent manner (Figure 3B). As SOX2 is a transcription factor and represses EnhII/Cp activity, we assessed the binding ability of SOX2 with EnhII/Cp. HepG2 cells and Huh7 cells were co-transfected with pGL-3-EnhII/Cp-Luc and pcDNA3.1-SOX2. Chromatin immunoprecipitation (ChIP) assay and qRT-PCR showed that SOX2 could recognize and bind to EnhII/Cp DNA in HepG2 cells (Figure 3C,D) and Huh7 cells (Figure 3E,F). More importantly, ChIP assay (Figure 3G) and qRT-PCR (Figure 3H) indicated that endogenous SOX2 could bind to EnhII/Cp DNA in HepAD38 cells. Taken together, these results revealed that SOX2 suppressed HBV replication at the transcriptional level by binding to EnhII/Cp and inhibiting the promoter activation.

### 3.4. The HMG Domain is Required for SOX2-Mediated Repression of HBV Replication

Next, the roles of different SOX2 protein domains in the suppression of HBV replication were evaluated. Plasmids carrying the truncated SOX2 genes, pcDNA3.1-SOX2 (1–333 nt), pcDNA3.1-SOX2 (1–540 nt), pcDNA3.1-SOX2 (1–600 nt), pcDNA3.1-SOX2 (121–954 nt), pcDNA3.1-SOX2 (334–954 nt), pcDNA3.1-SOX2 (541–954 nt), and pcDNA3.1-SOX2 (601–954 nt) were generated (Figure 4A). HepG2 cells were co-transfected with pGL-3-EnhII/Cp-Luc and pcDNA3.1, pcDNA3.1-SOX2, or plasmids expressing the truncated proteins, respectively. The results from the dual-luciferase reporter assay showed that the activity of EnhII/Cp was significantly inhibited by SOX2, SOX2 (1–540 nt), SOX2 (1–600 nt), and SOX2 (121–954 nt) (Figure 4B, lanes 2, and 4–6), attenuated by SOX2 (1–333 nt) (Figure 4B, lane 3), but not affected by SOX2 (334–954 nt), SOX2 (541–954 nt), and SOX2 (601–954 nt) (Figure 4B, lanes 7–9). These results indicated that the sequences from 121 nt to 333 nt of SOX2 containing the high mobility group (HMG) domain were required for SOX2-mediated repression of EnhII/Cp activity. 

The roles of the HMG domain and the transcription activation (TA) domain of SOX2 in the repression of EnhII/Cp activity were further determined. HepG2 cells and Huh7 cells were co-transfected with pBlue-HBV1.3(D) along with pcDNA3.1, pcDNA3.1-SOX2, pcDNA3.1-SOX2 ΔHMG, or pcDNA3.1-SOX2 ΔTA, respectively. The Northern bolt showed that the levels of HBV RNAs, 3.5 kb RNA, 2.4 kb RNA, and 2.1 kb RNA, were significantly reduced by SOX2 and SOX2 ΔTA, but not affected by SOX2 ΔHMG in HepG2 cells (Figure 4C) and Huh7 cells (Figure 4D). Similarly, qRT-PCR indicated that the levels of HBV core-associated DNA were significantly decreased by SOX2 and SOX2 ΔTA, but relatively unaffected by SOX2 ΔHMG in HepG2 cells (Figure 4E) and Huh7 cells (Figure 4F). In addition, the ELISA assay revealed that the levels of HBsAg and HBeAg were significantly reduced by SOX2 and SOX2 ΔTA, but relatively unaffected by SOX2 ΔHMG in HepG2 cells (Figure 4G) and Huh7 cells (Figure 4H). Therefore, these data suggested that the HMG domain was involved in SOX2-mediated repression of HBV replication. 

Moreover, the ChIP assay revealed that SOX2 and SOX2 ΔTA could bind to EnhII/Cp promoter DNA, but SOX2 ΔHMG failed to bind to EnhII/Cp DNA in HepG2 cells (Figure 4I) and Huh7 cells (Figure 4J). Furthermore, the results from the nuclear and cytoplasmic extraction analysis showed that in Huh7 cells, SOX2 and SOX2 ΔTA were detected in the whole cell lysates (Figure 4K, lanes 2 and 4), the nucleus (Figure 4K, lanes 6 and 8), and the cytoplasm (Figure 4K, lanes 10 and 12), while SOX2 ΔHMG was detected in the whole cell lysates (Figure 4K, lane 3) and the cytoplasm (Figure 4K, lane 11), but not in the nucleus (Figure 4K, lane 7), suggesting that SOX2 ΔHMG failed to enter the nucleus. Finally, immunofluorescence results revealed that SOX2 and SOX2 ΔTA were localized in both the cell nucleus and cytoplasm, while SOX2 ΔHMG was localized exclusively in the cytoplasm but not in the nucleus (Figure 4L), confirming that SOX2 ΔHMG failed to enter the nucleus. Collectively, these results demonstrated that the HMG domain of SOX2 was required for the anti-HBV activity of SOX2.

### 3.5. SOX2 Suppresses HBV Replication in BALB/c Mice 

The effect of SOX2 on HBV replication was further investigated in a mouse model. BALB/c mice (5-week-old, eight for each group) were co-injected with pAAV-HBV1.3 and pcDNA3.1, pcDNA3.1-SOX2, or pcDNA3.1-SOX2 ΔHMG through hydrodynamic injection. The blood was collected at 2 days and 4 days post-injection, and the levels of HBsAg and HBeAg in mice sera were measured by ELISA. The results revealed that the levels of HBsAg and HBeAg were significantly reduced by SOX2, but not by SOX2 ΔHMG (Figure 5A,B), suggesting that SOX2 represses the secretion of HBsAg and HBeAg in mice and HMG domain is required for such repression. Similarly, the level of HBV 3.5 kb RNA was significantly decreased by SOX2, but relatively unaffected by SOX2 ΔHMG (Figure 5C), indicating that SOX2 attenuates the expression of HBV 3.5 kb RNA in mice and HMG domain is required for such suppression. Moreover, immunohistochemical (IHC) staining of the mice liver tissues showed that the level of hepatitis B virus core protein (HBc) was significantly reduced (Figure 5(Db) vs. Figure 5(Da)) in the presence of SOX2 (Figure 5(Dd,De)), but relatively unchanged (Figure 5(Dc) vs. Figure 5(Da)) in the presence of SOX2 ΔHMG (Figure 5Dd,Df). Hematoxylin and eosin (H&E) staining showed that there was no severe pathological liver damage, including inflammation in livers treated with HBV and pcDNA3.1, SOX2, or SOX2 ΔHMG (Figure 5D). Taken together, our data revealed that SOX2 suppressed HBV gene expression and replication and the HMG domain was required for SOX2-mediated suppression of HBV replication in vivo.

## 4. Discussion

This study demonstrated the correlation between HBV replication and SOX2 expression. HBV activates SOX2 expression in human hepatoma cells and SOX2, in turn, represses HBV replication. In the previous reports, SOX2 expression was induced by HBV viral protein HBx. These findings matched the results of our finding that the expression level of SOX2 in HBV-positive HCC tissues was much higher as compared to that in healthy livers. Researchers also found that SOX2 expression was frequently over-expressed in dysplasia and HCV-associated HCC tissues, compared to non-tumor tissues [20]. Collectively, previous reports and our results revealed that over-expression of SOX2 may lead to the development of liver diseases. Induced-SOX2, in turn, represses the levels of HBV RNAs, HBV core-associated DNA, HBsAg, and HBeAg in dose-dependent manners, thereby suppressing HBV replication in HepG2 cells and Huh7 cells. More interestingly, the suppressive effects of SOX2 on HBV RNA transcription and HBV protein production are also confirmed in the BALB/c mice sera and liver tissues. We suggested that SOX2 may act as a restriction factor to repress HBV replication, not only during the acute infection, but also in the chronic infection and the development of liver diseases [12,20]. Previous studies have reported that HBV mRNA was lower in the HCC tissues relative to the non-tumor liver tissues [21,22]. In this study, we showed that the level of SOX2 was significantly higher in the HBV-positive HCC tissues. The mechanism of SOX2-supressed HBV replication was further investigated. SOX2 inhibits the activities of HBV EnhII/Cp, and both overexpressed SOX2 and endogenous SOX2 can bind to the EnhII/Cp DNA. Therefore, SOX2 represses HBV replication through binding to EnhII/Cp DNA and suppressing EnhII/Cp activity. Enhance II and core promoter are essential for the transcriptional regulation of HBV gene expression. Members of host transcriptional factors affect the activity of EnhII/Cp, such as liver-enriched hepatocyte nuclear factor (HNF), fetoprotein transcription factor (FTF), SP1, and ZEB2 [23,24,25]. HNF1 suppresses EnhII/CP activity via directly binding to EnhII [26], while HNF3 and HNF4 induce HBV transcription via directly binds to EnhII/CP [27,28]. FTF activates the core promoter and is the most efficient partners with HNF4a for the co-activation of EnhII/Cp [29]. SP1 binds to GC-rich DNA sequences on HBV enhancers and activates the activity of EnhII [24]. As a transcription factor, ZEB2 can bind to HBV core promoter and inhibit its activity [25]. SOX2 inhibits the activities of HBV EnhII/Cp perhaps through blocking transcriptional coactivators or recruiting transcriptional inhibiting factors. We previously reported that HBV replication and SOX4 expression were tightly controlled by a positive feedback mechanism [30]. In this study, we revealed that SOX2 and HBV replication were controlled by a negative feedback mechanism. Here, we found a previously unknown function of the transcription factor SOX2 in the repression of EnhII/Cp activity. 

SOX2 encodes 317 amino acids and contains two domains: An HMG domain and a TA domain. The HMG domain, which is a highly conserved DNA-binding domain of SOX family proteins, contains nearly 80 amino acids [31]. This work revealed that sequence from 121 nt to 333 nt containing the HMG domain was required for SOX2-mediated repression of EnhII/Cp activity. The HMG domain contains a DNA-binding sequence and a nuclear localization signal (NLS) [32]. Detailed studies revealed that SOX2 ΔHMG, a truncated SOX2 lacking the HMG region, could not repress HBV replication, lost the ability to bind to EnhII/Cp DNA, and failed to enter the nucleus, further confirming a critical role of SOX2 HMG domain in SOX2-mediated repression of HBV replication. Notably, the HMG domain is required for SOX2- mediated suppression of HBV replication in mice. In conclusion, the above facts indicate a distinct mechanism underlying the role of SOX2 in the repression of HBV replication, and suggest that SOX2 acts as a potential new host restriction factor during HBV replication.

## Figures and Tables

**Figure 1 viruses-12-00273-f001:**
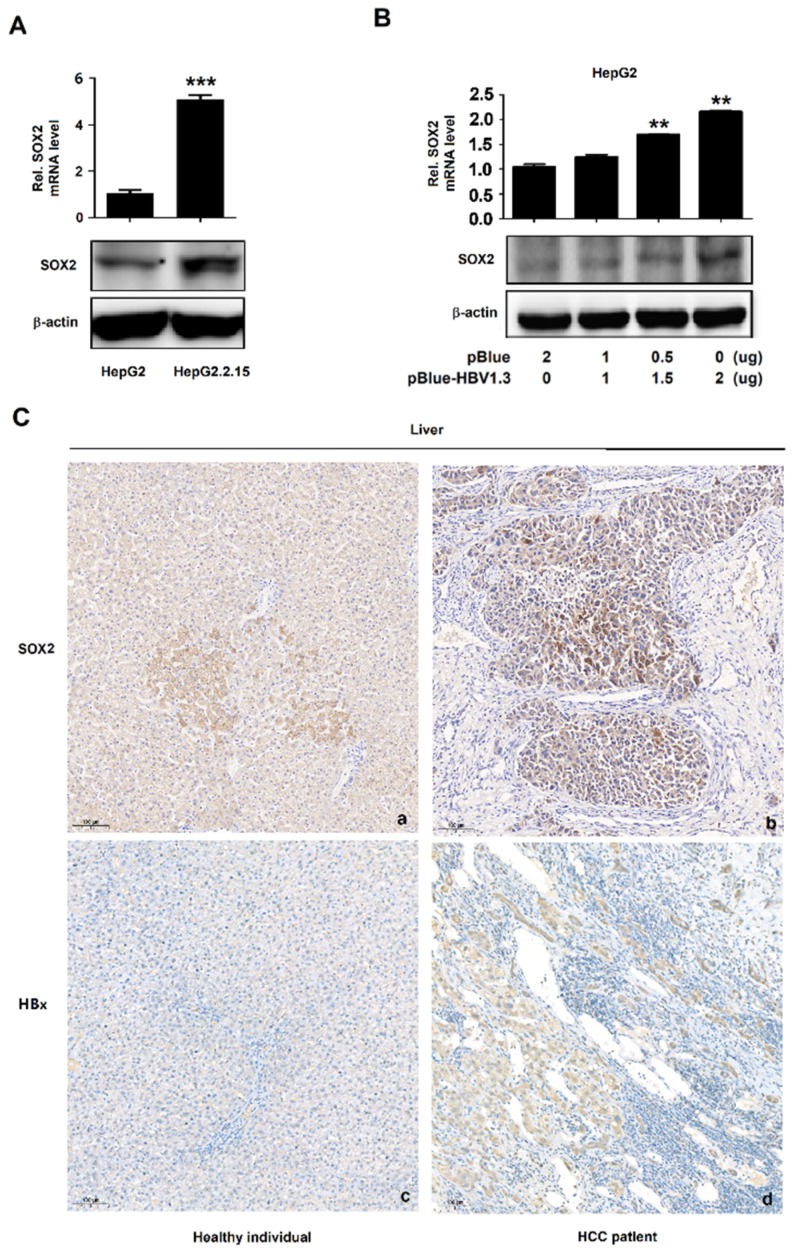
HBV induces sex determining region Y box2 (SOX2) expression. (**A**) The mRNA of SOX2 in HepG2 and HepG2.2.15 was determined by qRT-PCR (upper). GAPDH was used as the internal control. The protein of SOX2 in HepG2 and HepG2.2.15 were detected by Western blot (lower). β-actin was used as the internal control in Western blot. (**B**) HepG2 cells were plated in 6-well plates and transfected with pBlue or pBlue-HBV1.3 (genotype D). The cells were collected at 48 h post-transfection, and the mRNA and protein of SOX2 were detected by qRT-PCR (upper) and Western blot (lower). (**C**) Immunohistochemical staining of SOX2 and HBxAg (ab39716, abcam, Cambridge, England) in healthy liver and hepatocellular carcinoma (HCC) with hepatitis B virus (HBV) infection. **, *p* < 0.01 and ***, *p* < 0.001.

**Figure 2 viruses-12-00273-f002:**
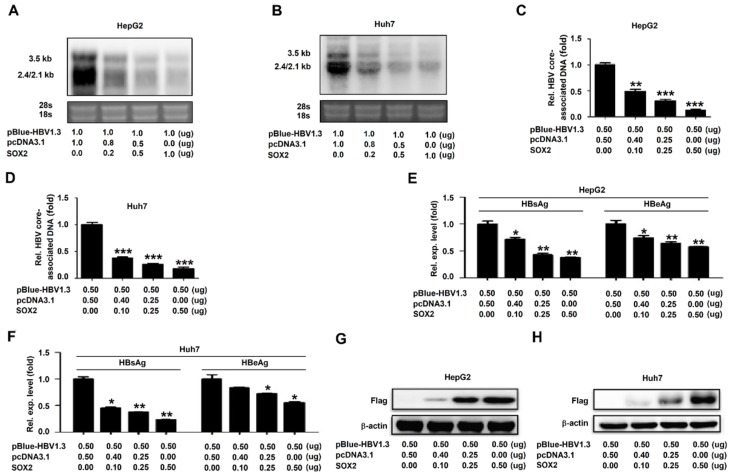
SOX2 represses HBV replication in HepG2 cells and Huh7 cells. (**A**–**H**) HepG2 cells (A, C, E, and G) and Huh7 cells (B, D, F, and H) were plated in 6-well plates and then co-transfected with pBlue-HBV1.3 and pcDNA3.1 or pcDNA3.1-SOX2 at different concentrations for 48 h. Total RNA was extracted and HBV RNAs were determined by Northern blot. The 28s and 18s rRNAs were used as the internal controls (A and B). HBV core-associated DNA was extracted and detected by qRT-PCR (C and D). HBsAg and HBeAg in cell culture supernatant were analyzed by ELISA (E and F). SOX2 and β-actin were detected by Western blot (G and H). *, *p* < 0.05, **, *p* < 0.01 and ***, *p* < 0.001.

**Figure 3 viruses-12-00273-f003:**
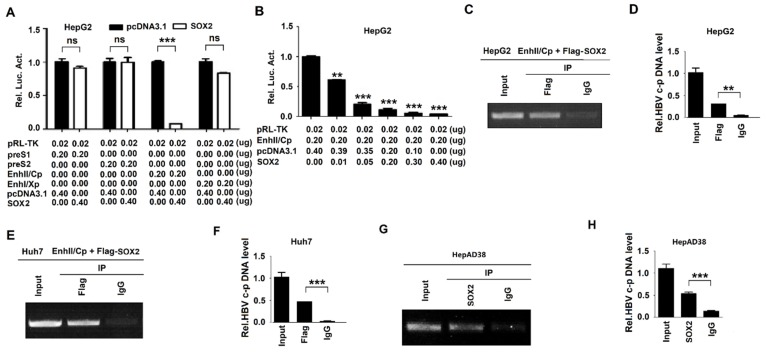
SOX2 represses HBV replication through inhibiting the activation of EnhII/Cp. (**A**) HepG2 cells were plated in 24-well plates and then co-transfected with pGL-3-preS1-Luc, pGL-3-preS2-Luc, pGL-3-EnhII/Cp-Luc, or pGL-3-EnhI/Xp-Luc along with pcDNA3.1 or pcDNA3.1-SOX2 for 48 h. pRL-TK was used as control for transfection efficiency. Luciferase activities of firefly relative to renilla were measured. (**B**) HepG2 cells were plated in 24-well plates and then co-transfected with pGL-3-EnhII/Cp-Luc and pcDNA3.1 or pcDNA3.1-SOX2 at different concentrations for 48 h. Luciferase activities were measured and normalized to the control. (**C**–**F**) HepG2 cells (C and D) and Huh7 cells (E and F) were plated in 6 cm dishes and co-transfected with 3 μg pGL-3-EnhII/Cp-Luc and 3 μg pcDNA3.1-SOX2 for 48 h. ChIP assays were performed using anti-Flag antibody to analyze the binding ability of SOX2 to EnhII/Cp. The extracted DNA was amplified by PCR and detected by agarose gel electrophoresis (C and E). The extracted DNA was detected by qRT-PCR (D and F). (**G** and **H**) HepAD38 cells were plated in 10 cm dishes and collected. ChIP assays were performed using anti-SOX2 antibody. The extracted DNA was amplified by PCR and detected by agarose gel electrophoresis (G). The extracted DNA was detected by qRT-PCR (H). NS, no significant (*p* > 0.05); *, *p* < 0.05, **, *p* < 0.01 and ***, *p* < 0.001.

**Figure 4 viruses-12-00273-f004:**
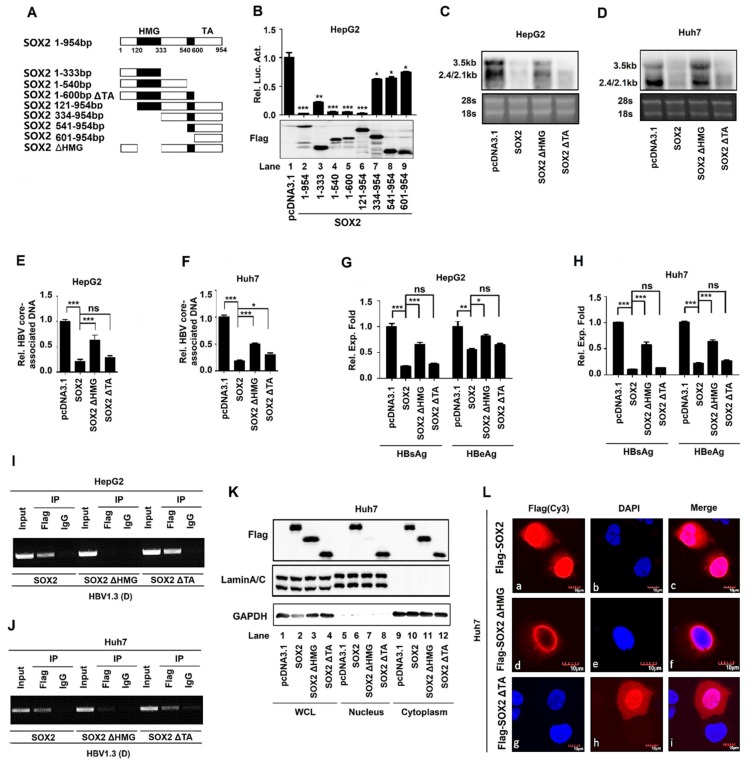
The HMG domain is required for SOX2-mediated repression of HBV replication. (**A**) Schematic diagrams of SOX2 functional domains and serial truncated mutants. (**B**) HepG2 cells were plated in 24-well plates and co-transfected with pGL-3-EnhII/Cp-Luc and pcDNA3.1, pcDNA3.1-SOX2, or pcDNA3.1-SOX2 (1–333 nt), pcDNA3.1-SOX2 (1–540 nt), pcDNA3.1-SOX2 (1–600 nt), pcDNA3.1-SOX2 (121–954 nt), pcDNA3.1-SOX2 (334–954 nt), pcDNA3.1-SOX2 (541–954 nt), or pcDNA3.1-SOX2 (601–954 nt), respectively, for 48 h. The activities of luciferase were measured and normalized to control (upper). The productions of SOX2 and its mutants were confirmed by Western blot (lower). (**C**,**D**) HepG2 cells (C) and Huh7 cells (D) were plated in 6-well plates and co-transfected with pBlue-HBV1.3 and pcDNA3.1, pcDNA3.1-SOX2, pcDNA3.1-SOX2 ΔHMG, or pcDNA3.1-SOX2 ΔTA, respectively, for 48 h. Total RNA was extracted and HBV 3.5 kb and 2.4/2.1 kb RNA were measured by Northern blot. The 28s and 18s rRNAs were used as the internal controls. (**E**–**H**) HepG2 cells (E and G) and Huh7 cells (F and H) were plated in 12-well plates and then co-transfected with pBlue-HBV1.3(D) and pcDNA3.1-SOX2, pcDNA3.1-SOX2 ΔHMG, or pcDNA3.1-SOX2 ΔTA, respectively, for 48 h. HBV core-associated DNA was extracted and detected by qRT-PCR (E and F). HBsAg and HBeAg in cell culture supernatants were analyzed by ELISA (G and H). (**I**,**J**) HepG2 and Huh7 cells were plated in 6 cm dishes and co-transfected with 3 μg pBlue-HBV1.3 and 3 μg pcDNA3.1, pcDNA3.1-SOX2, pcDNA3.1-SOX2 ΔHMG, or pcDNA3.1-SOX2 ΔTA, respectively, for 48 h. The cells were collected and ChIP assays were performed using anti-Flag antibody to detect the binding of SOX2, SOX2 ΔHMG, and SOX2 ΔTA to HBV DNA. (**K**) Huh7 cells were plated in 6-well plates and transfected with pcDNA3.1, pcDNA3.1-SOX2, pcDNA3.1-SOX2 ΔHMG, or pcDNA3.1-SOX2 ΔTA, respectively, for 48 h. Nuclear and Cytoplasmic Extraction Reagents were used to detect subcellular localization of SOX2 and its deletion mutants. LaminA/C and GAPDH were used as the internal controls in Western blot. (**L**) Huh7 cells were transfected with pcDNA3.1-SOX2, pcDNA3.1-SOX2 ΔHMG, and pcDNA3.1-SOX2 ΔTA, respectively, for 48 h. The subcellular localizations of SOX2, SOX2 ΔHMG, and SOX2 ΔTA were determined by immunofluorescence staining. The data was analyzed through One-way ANOVA and Tukey: Compare all pairs of columns. NS, no significant (*p* > 0.05); *, *p* < 0.05, **, *p* < 0.01 and ***, *p* < 0.001.

**Figure 5 viruses-12-00273-f005:**
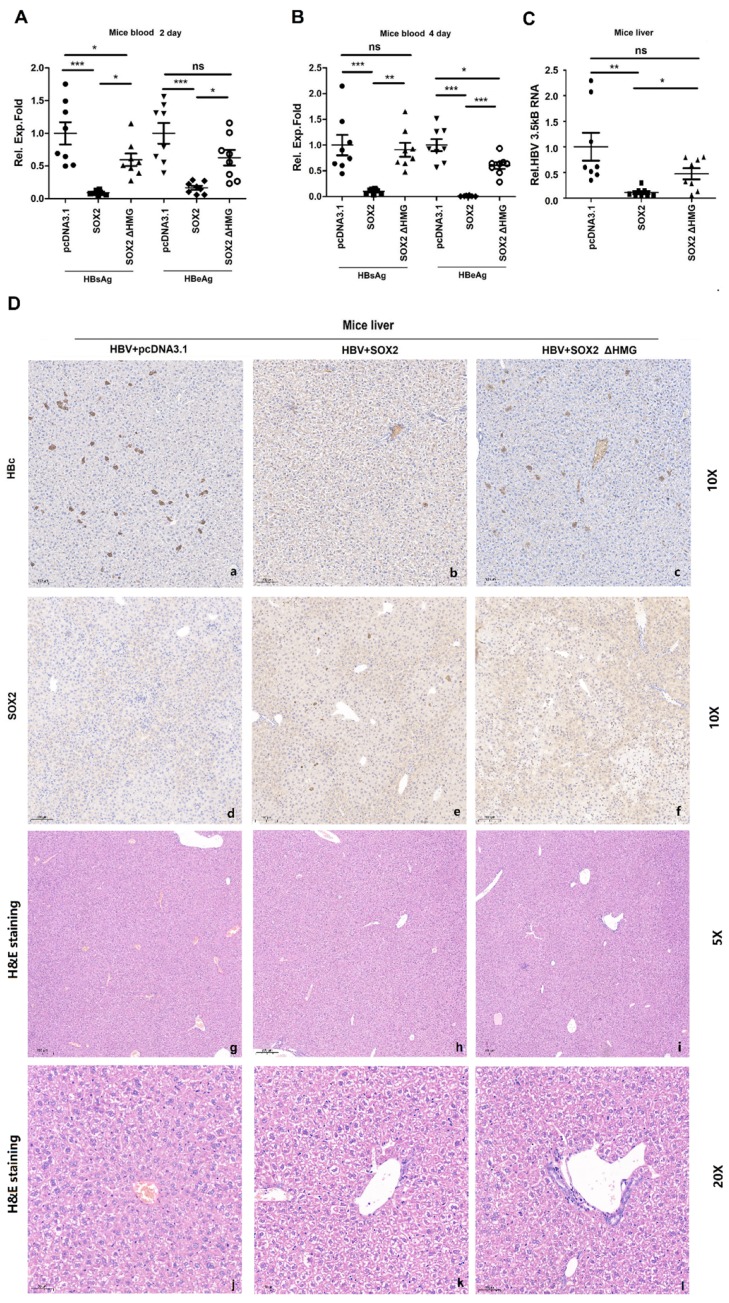
SOX2 suppresses HBV replication in BALB/c mice. (**A**–**D**) Three groups of BALB/c mice (5-week old, eight for each group) were co-injected with pAAV-HBV1.3 along with pcDNA3.1, pcDNA3.1-SOX2, or pcDNA3.1-SOX2 ΔHMG. Secreted HBsAg and HBeAg expressed in the mice blood were measured by ELISA at 2 days (A) and 4 days (B) post-injection. The mice were sacrificed and liver tissues were collected at 4 days post-injection. Total RNA was extracted from the liver tissues and the levels of HBV 3.5 kb mRNAs were determined by qRT-PCR (C). The abundances of HBcAg, SOX2, and SOX2 ΔHMG proteins in the liver tissues were detected by immunohistochemical staining (D). Data were analyzed through One-way ANOVO and Tukey: Compare all pairs of columns. NS, no significant (*p* > 0.05); *, *p* < 0.05, **, *p* < 0.01 and ***, *p* < 0.001.

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
