# Peer review of "SOX2 Represses Hepatitis B Virus Replication by Binding to the Viral EnhII/Cp and Inhibiting the Promoter Activation"

_viruses, 2020, doi:10.3390/v12030273_

Round 1

Reviewer 1 Report

Authors describe a host protein, SOX2 that when in high levels can suppress HBV replication. Overall an interesting paper, with nicely designed cell culture studies and assessments, but there are some areas that would need improvement, and I do have some design concerns.

-further description of the natural levels of SOX2 in the cell (if known) would be a great addition to the introduction. It appears this protein has many functions in replication, nearly all for supporting it. Is there a link that can be made with something that stimulates levels in HBV? In addition to introduction, this could be greatly expanded upon in the discussion--are levels of SOX2 being suppressed by HBV? is there something about that binding region that's comparable to other inhibitors in the literature?

-How do the SOX2 levels produced in your models compare with natural levels, in say, non-infected/transfected cells? I don't see any control cells in the experiments** Is HBV suppressing this?

-Your mouse model used is BALB/c. Have any modifications been made to this mouse? I was not aware that HBV replication could be supported in this mouse model, as they naturally don't have the NTCP cell receptors that are needed for HBV entry. Unless you're implying that all the replication that's occurring is in the plasma/PBMCs which might be possible. But how are you explaining your liver results?? I'm not sure what the liver stains are showing at that magnification, then. There is no reference and very little description of the method used here. 

-Multiple typos and errors in English grammar

Author Response

Response to Reviewer 1 Comments

Authors describe a host protein, SOX2 that when in high levels can suppress HBV replication. Overall an interesting paper, with nicely designed cell culture studies and assessments, but there are some areas that would need improvement, and I do have some design concerns

Point 1: -further description of the natural levels of SOX2 in the cell (if known) would be a great addition to the introduction. It appears this protein has many functions in replication, nearly all for supporting it. Is there a link that can be made with something that stimulates levels in HBV? In addition to introduction, this could be greatly expanded upon in the discussion--are levels of SOX2 being suppressed by HBV? is there something about that binding region that's comparable to other inhibitors in the literature?

Authors’ Response: Thank you very much for your comments. In the previous study (PMID: 27925189 and 28339084) and in our study (Revised Fig. 1A, B and C), the level of SOX2 was low in the normal cells and liver tissue, HBx which is one of HBV protein could induce SOX2 expression in cells. SOX2 inhibits the activities of HBV Enhâ…¡/Cp perhaps through blocking transcriptional coactivators or recruiting transcriptional inhibiting factor. The length of Enhâ…¡/Cp nearly is 260bp and many transcriptional factor inhibits the activities of HBV Enhâ…¡/Cp such as ZEB2 and HNF1,we could not confirm the binding region of SOX2 which was comparable to other inhibitors in the literature.

Point 2:-How do the SOX2 levels produced in your models compare with natural levels, in say, non-infected/transfected cells? I don't see any control cells in the experiments** Is HBV suppressing this?

Authors’ Response: Thank you for the comments. The expression level of SOX2 was low in HepG2 and induced by HBV (Revised Fig. 1A and B).

Point 3:-Your mouse model used is BALB/c. Have any modifications been made to this mouse? I was not aware that HBV replication could be supported in this mouse model, as they naturally don't have the NTCP cell receptors that are needed for HBV entry. Unless you're implying that all the replication that's occurring is in the plasma/PBMCs which might be possible. But how are you explaining your liver results?? I'm not sure what the liver stains are showing at that magnification, then. There is no reference and very little description of the method used here. 

Authors’ Response: Thank you very much for your comments. Our mouse model used is BALB/c and do not have any modifications been made in this mouse. Although BALB/c mouse do not have NTCP cell receptors and could not infect by HBV directly, but through hydrodynamic injection (HI), HBV could produce HBsAg and HBeAg in the serum as well as HBcAg expression in liver tissue (PMID: 29070073).This HBV hydrodynamic injection (HI) mouse model was used in our lab and another lab before (PMID: 30674631 and 25653429).

Point 4:-Multiple typos and errors in English grammar

Authors’ Response: Thank you. Multiple typos and errors in English grammar were corrected.

Reviewer 2 Report

Manuscript “SOX2 represses Hepatitis B Virus replication by binding to the viral EnhII/Cp and inhibiting promotor activation”

Comments

Authors have conducted the extensive in vitro and in vivo experiments to document another pathway of repression of HBV replication. Their study showed that SOX2, a transcription factor, containing HMG domain binds to the viral Enh II/Cp and inhibit the promotor activation that resulted in repression of HBV replication. Their work is commendable. It is hoped that this work would lead to further development for the treatment of HBV infection.

Minor comments

Several typos and grammatical errors

Author Response

Response to Reviewer 2 Comments

Authors have conducted the extensive in vitro and in vivo experiments to document another pathway of repression of HBV replication. Their study showed that SOX2, a transcription factor, containing HMG domain binds to the viral Enh II/Cp and inhibit the promotor activation that resulted in repression of HBV replication. Their work is commendable. It is hoped that this work would lead to further development for the treatment of HBV infection.

Minor comments

Several typos and grammatical errors

Authors’ Response: Thank you. Several typos and grammatical errors were corrected.

Reviewer 3 Report

In this manuscript Yang et al. present evidence that the Sox2 protein binds to, and represses, the HBV core promoter. This observation is validated in a variety of cell types and in a mouse model system. In general, the experiments are presented in a logical sequence and the data appear to support the findings of the paper.

I do have some concerns about the quality of the manuscript and the amount of technical detail provided, as listed below.

Specific comments:

English is generally poor. While the flow of the experiments can be followed, the text itself is littered with errors that should be corrected, probably by a professional editing service. Methods require more detail, including: 2.2 - first and last sentences are duplicated. Is there a citation for the plasmids from Prof. Guo? 2.3 - More description of the cell types used would help. Also, seeding densities? 2.4 - How much RNA used for cDNA synthesis? How much cDNA used for qPCR? 2.5 - What antibody concentrations were used? What type of ECL? 2.6 - Even if the details are in another paper, it would be nice to have a brief summary of the general method here. 2.7 - 28S and 18S ribosomal RNAs? The authors should clarify what they are and what role they play here. 2.8 - What exactly is being repeated? The whole assay from seeding cells onwards, or just the luciferase reading? 2.11 - Antibody dilutions, washing, mounting media? What type of objective was used. Were the images adjusted prior to submission? 2.12 - In some cases there are more than 2 groups (e.g. Fig. 3), ANOVA + Tukey would be more appropriate. Also, some clarification about the nature of the replicates would be appreciated. Fig. 1 - A little more detail about how the graphed values are derived would help. It looks like Fluc is expressed relative to Rluc, and then everything relative to the average of the 0 ug SOX2 sample. Fig. 2 - Plasmid ratios for panel A are needed, and ideally panels C and D would have accompanying RT-qPCR data, as for E. Fig. 4D - scale bars and more detail on the immunohistochemistry would be good. These are H and E stained sections? There doesn’t seem to be a corresponding method Discussion - The discussion could be extended to better explain the relationship between the new data and the pre-existing knowledge in the field.  

Author Response

Response to Reviewer 3 Comments

In this manuscript Yang et al. present evidence that the Sox2 protein binds to, and represses, the HBV core promoter. This observation is validated in a variety of cell types and in a mouse model system. In general, the experiments are presented in a logical sequence and the data appear to support the findings of the paper.

I do have some concerns about the quality of the manuscript and the amount of technical detail provided, as listed below.

Specific comments:

English is generally poor. While the flow of the experiments can be followed, the text itself is littered with errors that should be corrected, probably by a professional editing service. Methods require more detail, including: 2.2 - first and last sentences are duplicated. Is there a citation for the plasmids from Prof. Guo? 2.3 - More description of the cell types used would help. Also, seeding densities? 2.4 - How much RNA used for cDNA synthesis? How much cDNA used for qPCR? 2.5 - What antibody concentrations were used? What type of ECL? 2.6 - Even if the details are in another paper, it would be nice to have a brief summary of the general method here. 2.7 - 28S and 18S ribosomal RNAs? The authors should clarify what they are and what role they play here. 2.8 - What exactly is being repeated? The whole assay from seeding cells onwards, or just the luciferase reading? 2.11 - Antibody dilutions, washing, mounting media? What type of objective was used. Were the images adjusted prior to submission? 2.12 - In some cases there are more than 2 groups (e.g. Fig. 3), ANOVA + Tukey would be more appropriate. Also, some clarification about the nature of the replicates would be appreciated. Fig. 1 - A little more detail about how the graphed values are derived would help. It looks like Fluc is expressed relative to Rluc, and then everything relative to the average of the 0 ug SOX2 sample. Fig. 2 - Plasmid ratios for panel A are needed, and ideally panels C and D would have accompanying RT-qPCR data, as for E. Fig. 4D - scale bars and more detail on the immunohistochemistry would be good. These are H and E stained sections? There doesn’t seem to be a corresponding method Discussion - The discussion could be extended to better explain the relationship between the new data and the pre-existing knowledge in the field. 

Authors’ Response: Thank you very much for your comments. Multiple typos and errors be corrected. Methods were added more detail.

2.2 - last sentences are deleted.

2.3 -The hepatocarcinoma cell lines HepG2, Huh7, HepG2.2.15 cells (derived from HepG2 cells carrying HBV genome) and HepAD38 (replicates HBV under conditions regulated with tetracycline). The cells were seeded in 24-well (seeding densities: 2.5×105), 12-well (seeding densities: 5×105), 6-well (seeding densities: 1.2×106), and 6cm vessel (seeding densities: 2.6×106).

2.4 –5μg RNA was used to synthesize cDNA by using random prime. 2μl cDNA was detected by a Light Cycler 480 instrument.

2.5 – The dilutions of primary antibodies and secondary antibody were added in the text. The proteins were detected with ECL (1705060 Bio-Rad American) by LAS-4000 image document instrument.

2.6 –A brief summary of HBV core-associated DNA purified were added in the text.

2.7 - 28S and 18S ribosomal RNAs was used as an internal control to measure the same amount of total RNA loading.

2.8 - The whole assay contains seeding cells and luciferase reading.

2.11-The dilution of flag is 1:100. The cells were stained with DAPI. The images were adjusted prior to submission.

2.12 ANOVA + Tukey was used to analysis the date of cells and mouse (Revised Fig. 4 and 5).

Dual-luciferase reporter assay was measured through Fluc relative to Rluc, and then everything relative to the average of the 0 ug SOX2 sample.

Plasmid ratios for panel A are added (Revised Fig. 3A).

qRT-PCR analysis was added to measure the concentration of extracted DNA (Revised Fig. 3D and F).

scale bars on the immunohistochemistry was added (Revised Fig. 1C and 5D). Hematoxylin and eosin (H&E) staining showed that there is no severe pathological

liver damage including inflammation in mouse liver treated with HBV and pcDNA3.1, SOX2 and SOX2 ΔHMG (Revised Figure 5D).

Round 2

Reviewer 1 Report

Overall, I think it's nice paper with a good design and interesting findings. The additional clarifications on methods, etc., helped many of the points that were lacking in the first draft. There are still parts of the text that I find are missing links/key information.

--In discussion, there needs to be more analysis of how this fits into the big picture. The SOX2 is at baseline, then cells get HBV and levels increase due to HBx, but is inhibitory to the virus. What does this imply? good target? bad target? just an observation?

--Was the amount of SOX2 induced in the models something physiological? or did it interfere with another pathway that overall affected health of the cells?

--Should be at least addressed, but using HepG2 and HepG2.2.15 is not the same as infecting(/transfecting) a virus into the same cell line. These cell lines have been growing separately in labs for years, potentially compounding their differences. I don't deny that the levels are likely different because of the HBV, but they're not necessarily a direct comparison. And both cell lines are of malignant origin, which you highlight as having abnormalities of SOX2 in cancers.  

--ZEB2 transcription factor sentence does not fit as is...why did you bring this up? what's the link with SOX2? There are lots of transcription factors that bind this region...(see few reviews/references)

--For the mouse studies, is there a normal liver control (ie, without HBV)?

Other:

-many typos throughout--I've highlighted a few in the file, but this needs to be reviewed again. 

-check your figure labelling--you have pcDNA3.1 and pCDNA1.3 mixed around in several spots.

-Promoter name SP1 and SP2 need to be changed to the literature accepted names: preS1 and preS2

Author Response

Responses to the Reviewers Overall, I think it's nice paper with a good design and interesting findings. The additional clarifications on methods, etc., helped many of the points that were lacking in the first draft. There are still parts of the text that I find are missing links/key information. Authors’ Response: Thank you for the comments. --In discussion, there needs to be more analysis of how this fits into the big picture. The SOX2 is at baseline, then cells get HBV and levels increase due to HBx, but is inhibitory to the virus. What does this imply? good target? bad target? just an observation? Authors’ Response: Thank you for the comments. As you suggested, we discussed the potential impact of SOX2 on HBV replication and disease development in the discussion section of the revised manuscript. “We suggest that SOX2 may act as a restriction factor to repress HBV replication not only during the acute infection but also in the chronic infection and the development of liver diseases [12, 20]. As previous studies have reported that HBV mRNA was lower in the HCC tissues relative to the non-tumor liver tissues [21, 22], and in this study, we showed that the level of SOX2 was significantly higher in the HBV-positive HCC tissues.”. --Was the amount of SOX2 induced in the models something physiological? or did it interfere with another pathway that overall affected health of the cells? Authors’ Response: Thank you for the comment. Previous studies reported that SOX2 is a main downstream regulator of the silent information regulator 1 (SIRT1)-mediated self-renewal and tumorigenicity potential of liver cancer stem cells (PMID:27312708), and SOX2 activates the expression of programmed death ligand-1 (PD-L1) through directly bound to the PD-L1 promoter. Binding of PD-L1 to its receptor, the programmed death-1 (PD-1), leads to tumor evasion from host immune system (PMID:28339084). --Should be at least addressed, but using HepG2 and HepG2.2.15 is not the same as infecting (/transfecting) a virus into the same cell line. These cell lines have been growing separately in labs for years, potentially compounding their differences. I don't deny that the levels are likely different because of the HBV, but they're not necessarily a direct comparison. And both cell lines are of malignant origin, which you highlight as having abnormalities of SOX2 in cancers. Authors’ Response: Thank you for the comments. Yes, you are right. Indeed, HepG2 and HepG2.2.15 have been growing separately in labs for years. HepG2.2.15 containing the complete HBV genome and capable of stable HBV replication and gene expression in the cell culture system. HepG2 and HepG2.2.15 cells have been used to compare the changing situation of gene expression (PMID 22610944 and PMID 27253403). As you suggested, we have discussed that SOX2 expression was induced in HepG2 transfected with HBV. --ZEB2 transcription factor sentence does not fit as is…why did you bring this up? What’s the link with SOX2? There are lots of transcription factors that bind this region…(see few reviews/references) Authors’ Response: Thank you for the question. Yes, several host transcriptional factors ca affect the activity of Enhâ…¡/Cp, such as hepatocyte nuclear factor (HNF), fetoprotein transcription factor (FTF), and ZEB2. As ZEB2 is a transcription factor can bind to HBV core promoter to inhibit its activity, we thus used it as example for the study. --For the mouse studies, is there a normal liver control (ie, without HBV)? Authors’ Response: Thank you for the question. No, in the mouse studies, there was no normal liver control (without HBV). Other: -many typos throughout—I’ve highlighted a few in the file, but this needs to be reviewed again. Authors’ Response: Thank you so much for the help. According to your suggestion, we have reviewed the manuscript and modified the errors. -check your figure labelling--you have pcDNA3.1 and pCDNA1.3 mixed around in several spots. Authors’ Response: Thank you for the comment. The correct one is pcDNA3.1. We have changed pcDNA3.1 to pCDNA1.3 in the revised manuscript. -Promoter name SP1 and SP2 need to be changed to the literature accepted names: preS1 and preS2 Authors’ Response: Thanks again. As you suggested, we have changed SP1 and SP2 to preS1 and preS2 in the revisions.
